# YAP drives cutaneous squamous cell carcinoma formation and progression

Zoé Vincent-Mistiaen[1†], Ahmed Elbediwy[1†], Hannah Vanyai[1], Jennifer Cotton[2], Gordon Stamp[1], Emma Nye[1], Bradley Spencer-Dene[1], Gareth J Thomas[3], Junhao Mao[2], Barry Thompson[1]*

[1]The Francis Crick Institute, London, United Kingdom; [2]University of Massachusetts Medical School, Worcester, United States; [3]Cancer Sciences Unit, Faculty of Medicine, University of Southampton, Southampton, United Kingdom

**Abstract** Squamous cell carcinoma (SCC) can progress to malignant metastatic cancer, including an aggressive subtype known as spindle cell carcinoma (spSCC). spSCC formation involves epithelial-to-mesenchymal transition (EMT), yet the molecular basis of this event remains unknown. The transcriptional co-activator YAP undergoes recurrent amplification in human SCC and overexpression of YAP drives SCC formation in mice. Here, we show that human spSCC tumours also feature strong nuclear localisation of YAP and overexpression of activated YAP (NLS-YAP-5SA) with Keratin-5 (K5-CreERt) is sufficient to induce rapid formation of both SCC and spSCC in mice. spSCC tumours arise at sites of epithelial scratch wounding, where tumour-initiating epithelial cells undergo EMT to generate spSCC. Expression of the EMT transcription factor ZEB1 arises upon wounding and is a defining characteristic of spSCC in mice and humans. Thus, the wound healing response synergises with YAP to drive metaplastic transformation of SCC to spSCC.
DOI: https://doi.org/10.7554/eLife.33304.001

*For correspondence:
barry.thompson@crick.ac.uk

†These authors contributed equally to this work

Competing interests: The authors declare that no competing interests exist.

## Introduction

Squamous cell carcinoma (SCC) is a type of epithelial cancer that is known to originate in the stratified squamous epithelia of the body, such as the skin, cervix, oesophagus, or oral cavity. Spindle cell carcinoma (spSCC) is a morphologically distinct type of cancer that consists of spindle-shaped non-epithelial cells, yet can occur in the same bodily locations as SCC. The cell type of origin for spSCC is thought to be epithelial, but is still not definitively established. Histological analysis of spindle cell carcinomas identified a 'sarcomatoid' morphology, resembling the mesenchymal fibroblasts of malignant fibrosarcomas, which suggests a possible mesodermal origin. However, the spindle-shaped tumour cells of spSCC may instead derive from epithelial cells of the epidermis via an epithelial-to-mesenchymal transition (EMT) (*Brabletz et al., 2018*; *Yang and Weinberg, 2008*), a process observed to occur in some mouse models of RasG12D-driven skin carcinomas (*Latil et al., 2017*). Decisive demonstration of the cell of origin of spSCC requires additional mouse models for spSCC that allow for tracing of cell lineages.

The molecular basis for spSCC formation remains a fundamental unsolved problem. Recent work has identified the YAP oncoprotein, originally discovered to drive cell proliferation in *Drosophila* (*Huang et al., 2005*), and later to drive liver tumours in mice (*Camargo et al., 2007*; *Dong et al., 2007*) and to be mechanically regulated (*Dupont et al., 2011*). In skin, YAP is a driver of epidermal cell proliferation in mouse embryos (*Zhang et al., 2011*) and SCC formation in embryonic mouse skin after transplantation into nude mice (*Schlegelmilch et al., 2011*). YAP knockouts also prevent Ras-driven skin SCC formation (*Debaugnies et al., 2018*; *Zanconato et al., 2015*). YAP is furthermore known to exhibit recurrent amplifications in human SCC tumours (*Hiemer et al., 2015*; *India Project Team of the International Cancer Genome Consortium, 2013*) and to promote

human SCC cell proliferation in culture (*Walko et al., 2017*). During normal skin development and homeostasis, YAP acts redundantly with its paralog TAZ to maintain proliferation of basal layer stem/progenitor cells and to promote wound healing (*Elbediwy et al., 2016*). Integrin-SRC signalling promotes YAP activity in cell culture, mouse skin and in human SCC cells in culture (*Elbediwy et al., 2016*; *Kim and Gumbiner, 2015*; *Li et al., 2016*). Thus, formation of SCC may occur simply by an acceleration of the normal YAP-dependent program of basal layer cell proliferation to produce an overgrown epidermis. In contrast, the molecular basis of SCC progression to spSCC remains poorly understood.

## Results and discussion

We sought to test whether YAP is involved in formation of spSCC as well as SCC. To address this question, we stained histological sections of normal human skin and human spSCC tumours with an anti-YAP antibody and an anti-Keratin-5 (K5) antibody to mark epidermal cells (*Figure 1A–C*). YAP is normally expressed primarily in the epidermis, and is nuclear localised in the basal layer stem progenitor cells (*Figure 1B,C*). In spSCC tumours, YAP is strongly expressed and nuclear localised throughout the tumour, despite the fact that the tumour is dermal rather than epidermal in character as revealed by the absence of K5 expression in the tumour (*Figure 1A–C*; *Figure 1—figure supplement 1*). We note that spSCC tumours are often associated with a wounded epidermis, as indicated by a gap in the K5-positive layer above the tumour. These results suggest that high levels of nuclear YAP and epidermal wounding may be involved in spSCC formation.

To test this notion, we generated a conditionally inducible YAP transgene (Rosa26 LoxSTOPLox NLS-YAP-5SA IRES LacZ) that can be activated by expression of the Cre recombinase enzyme. We included a lineage tracer (LacZ) to identify all daughter cells deriving from mother cells undergoing Cre-mediated recombination. Crossing this transgene with a Keratin-5 driven, tamoxifen-inducible, Cre recombinase line (K5-CreERt) enabled conditional expression of oncogenic NLS-YAP-5SA in skin epidermis after treatment with tamoxifen to induce nuclear localisation of the CreERt enzyme and excision of the PolyA-containing STOP cassette (*Figure 2A*). We find that NLS-YAP-5SA is able to induce formation of both SCC and spSCC in skin within 2–4 weeks of tamoxifen treatment (*Figure 2B*). SCC and spSCC can be distinguished because spSCC tumour cells are dermally located and lack expression of K5 (*Figure 2B*). Notably, spSCC tumours tended to arise in regions where the mice scratch their skin, and spSCC tumours were often associated with a wounded epidermis, while SCCs could arise in the absence of epidermal wounding (*Figure 2B*). These findings raise the question of how expression of oncogenic YAP in the K5-positive epidermal cells can give rise to K5-negative dermal spSCC, and whether wounding of the skin has a role in the process.

To follow the lineage of spSCC cells, we stained NLS-YAP-5SA expressing mouse skin for nuclear beta-Galactosidase (the product of the *LacZ* gene), which becomes expressed upon K5-CreERt-mediated recombination. Following treatment with tamoxifen for 5 days, nuclear beta-Gal expression was detected in patches of epidermal cells in skin areas not subject to scratch wounds (*Figure 2C*). In regions subject to scratching, large spSCCs formed with clear nuclear beta-Gal expression in all tumour cells and some neighbouring epidermal cells (*Figure 2C*). The spSCC tumours were highly proliferative, as indicated by Ki67-positive immunostaining (*Figure 2D,E*). These results demonstrate that oncogenic YAP drives spSCC formation by metaplastic transformation of K5-positive epidermal basal layer epithelial cells into K5-negative dermal mesenchymal spindle cells.

To explore the mechanism of metaplastic transformation during oncogenic YAP-driven spSCC formation, we examined the expression of classic epithelial-to-mesenchymal transition (EMT) markers (*Yang and Weinberg, 2008*). A defining feature of EMT is the downregulation of epithelial markers such E-cadherin or Keratins and upregulation of Vimentin. In nuclear YAP-driven tumours, Keratin-5 expression is inactivated and Vimentin expression is strongly induced upon transformation of SCC to spSCC following wounding (*Figure 3A,B*). EMT is known to be driven by E-box DNA binding transcription factors such as Twist (mammalian TWIST1), Snail (mammalian SNAI1 and SNAI2/SLUG), and Zfh1/2 (mammalian ZEB1/2) – all originally discovered in *Drosophila*, and then in other model organisms, to promote mesodermal rather than ectodermal fate during developmental EMT (*Boulay et al., 1987*; *Broihier et al., 1998*; *Fortini et al., 1991*; *Lai et al., 1991*, *1993*; *Leptin, 1991*; *Nieto et al., 1992*, *1994*; *Thisse et al., 1988*). We chose to focus on ZEB1 because (1) in *Drosophila*,

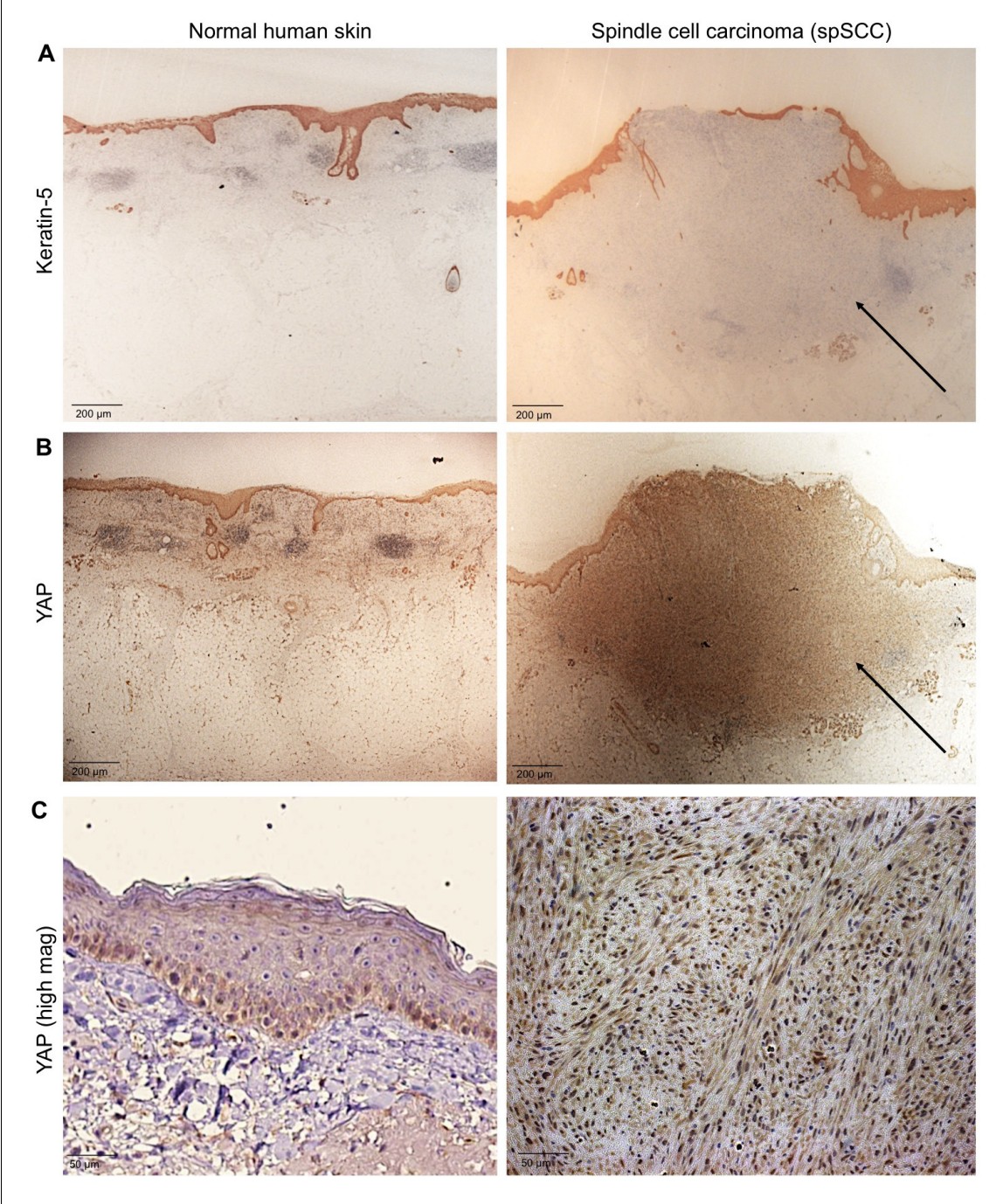

**Figure 1.** YAP is nuclear localised in human spindle cell carcinoma. (**A**) Histological sections of normal human skin and spindle cell carcinoma patient tumour stained for the epithelial marker Keratin-5 (brown immunostain). Scale bar 200 µM. (**B**) Histological sections of normal human skin and spindle cell carcinoma patient tumour stained for YAP (brown immunostain). Scale bar 200 µM. (**C**) High magnification view of (**B**) showing nuclear localisation of YAP protein in spindle cell carcinoma (brown immunostain). Sections are co-stained for eosin (blue). Scale bar 200 µM.

DOI: https://doi.org/10.7554/eLife.33304.002

The following figure supplement is available for figure 1:

**Figure supplement 1.** A panel of human spSCC tumours are characterised by widespread nuclear YAP localisation.

DOI: https://doi.org/10.7554/eLife.33304.003

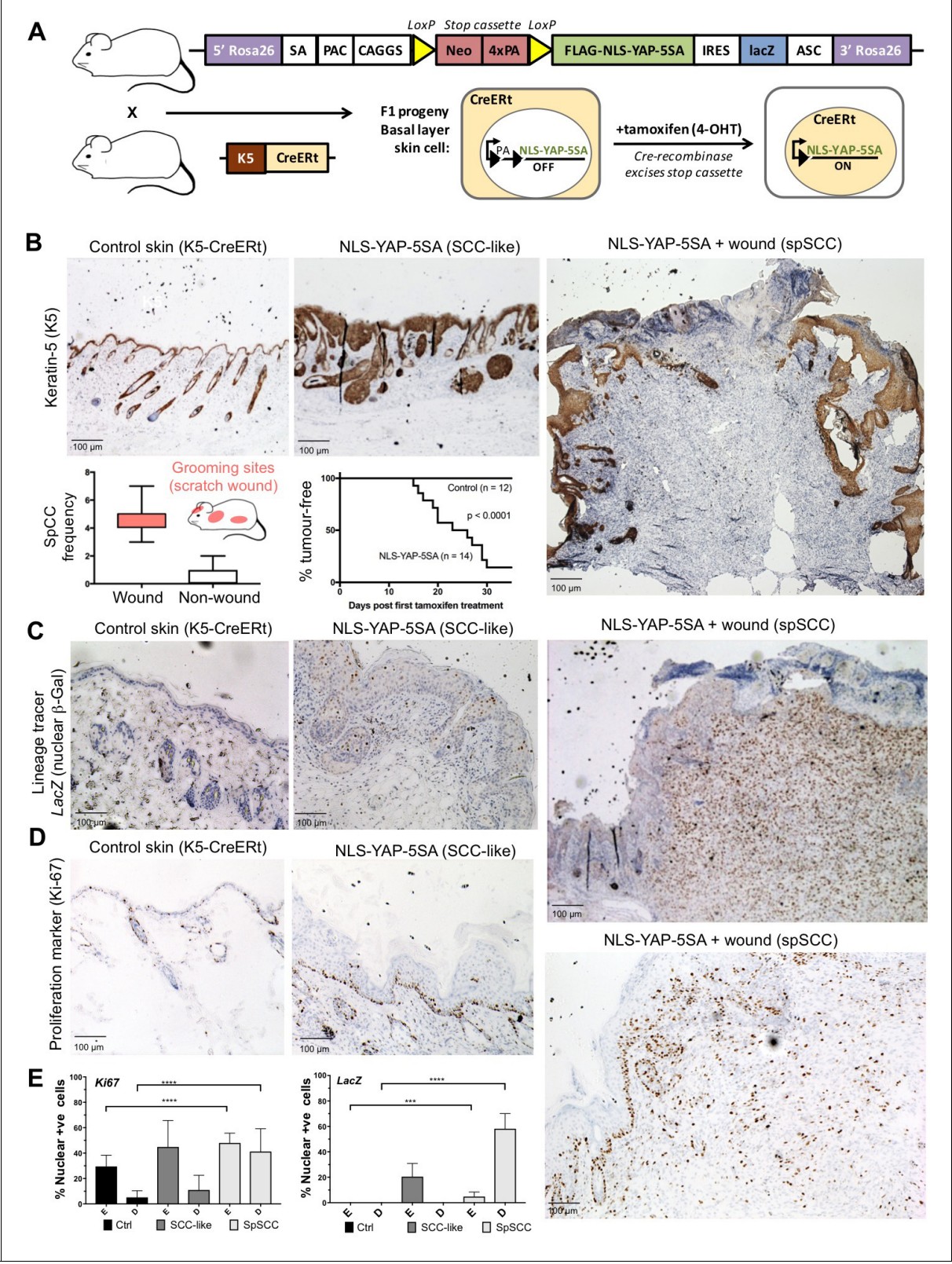

**Figure 2.** Nuclear YAP drives formation of both SCC and spSCC in mice. (**A**) Skin-specific expresison of nuclear YAP was achieved by crossing K5-CreERt mice to a Lox-Stop-Lox cassette for conditional expression of nuclear YAP and a LacZ lineage tracer in basal layer skin cells. (**B**) Expression of nuclear YAP drives formation of both SCC-like overgrowths (Keratin-5 positive) and spSCC-like tumours (mostly Keratin-5 negative). Multiple spSCC tumours arise per animal, but only in areas subject to scratch wounding. Note the disruption in the continuity of Keratin-5 positive epithelial layer above

*Figure 2 continued on next page*

*Figure 2 continued*

the spSCC tumour, indicative of a wound-induced tumour (*n* = 20). Kaplan-Meier analysis shows rapid induction of tumours in NLS-YAP-5SA expressing skin. (**C**) Lineage tracing with LacZ (encoding nuclear beta-Gal immunostained in brown) induced with the K5-CreERt line indicates that both SCC and spSCC tumours arise from the K5-positive basal layer of the skin (*n* = 22). (**D**) Proliferation of cells was measured by staining for the mitotic marker Ki-67 in control, SCC and spSCC samples (*n* = 25). Scale bars 100 µM. (**E**) Quantification of C and D in epidermal (**E**) vs dermal (**D**) compartments.
DOI: https://doi.org/10.7554/eLife.33304.004

Zfh1/2 are expressed downstream of Twist and Snail (*Lai et al., 1991*), (2) in mammalian cells in culture, ZEB1 (also called deltaEF1) and ZEB2 (also called SIP1) are known to be potent inducers of EMT (*Aigner et al., 2007*; *Comijn et al., 2001*; *Eger et al., 2005*; *Krebs et al., 2017*) and (3) ZEB1 was recently shown to cooperate with YAP to drive a common set of target genes in cell culture (*Lehmann et al., 2016*). In vivo, we find that ZEB1 mRNA and protein is normally not expressed in epithelial cells or SCC but becomes strongly expressed in oncogenic YAP-driven spSCC tumour cells, similar to SNAIL1 mRNA (*Figure 3C–F*). These results indicate that YAP-driven spSCC arises via an EMT program involving expression of ZEB1.

Importantly, we find that ZEB1 is also induced during the wound healing response in normal skin epithelia and in cultured skin keratinocytes (*Figure 4A,B*). Since the wound-induced upregulation of YAP and ZEB1 can be recapitulated in cultured keratinocytes, we examined whether the induction of ZEB1 expression that occurs during wound healing requires active YAP. To inhibit YAP activation, we treated scratch-wounded keratinocyte monolayers with either siRNAs targeting YAP/TAZ or with the Src family kinase inhibitor Dasatinib, which is known to strongly inhibit YAP activation (*Elbediwy et al., 2016*; *Kim and Gumbiner, 2015*; *Li et al., 2016*). We find that inhibition of YAP activity with this compound prevents activation of ZEB1 expression at 4 hr after scratch wounding (*Figure 4C,D*). These observations indicate that wound-induced ZEB1 expression is normally sustained by YAP activity. To confirm that YAP acts via TEAD-dependent transcription, we silenced expression of TEAD1-4 with siRNAs, which prevented ZEB1 induction after scratch wounding (*Figure 4E*). Finally, we were able to detect TEAD1 binding to an upstream enhancer at the *ZEB1* locus upon scratch wounding, although the degree of enrichment is weak due to the very small percentage of *ZEB1* expressing cells in this experiment (*Figure 4F*). Importantly, the physiological upregulation of YAP and ZEB1 following wounding is only transient, as these genes are normally silenced following wound closure, which inhibits YAP. In contrast, ZEB1 expression is not silenced in the presence of oncogenic NLS-YAP-5SA, allowing ZEB1 to drive EMT and spSCC formation (*Figure 3A–C*). Finally, we sought to confirm that our findings with transgenic mice also apply to humans. We first confirmed that ZEB1 becomes co-expressed with YAP and that there is a loss of E-cadherin and gain of Vimentin expression in human spSCC tumours (*Figure 5A–D*; *Figure 5—figure supplement 1*).

In conclusion, our findings identify YAP as a driver of both SCC and spSCC formation, demonstrate that both SCC and spSCC originate in the basal layer of the epidermis, and reveal that the SCC versus spSCC outcome is determined by whether or not tumour cells undergo a ZEB1-mediated EMT program that can be induced by skin wounding (*Figure 5E*). Thus, while SCC reflects an abnormal hyperactivation of homeostatic epidermal proliferation mechanisms (requiring YAP/TAZ (*Debaugnies et al., 2018*) but not ZEB1, which is not expressed in SCC), spSCC reflects an abnormal hyperactivation of wound repair proliferation and EMT mechanisms (driven by both YAP and ZEB1). Future work should aim to examine genetic deletion of ZEB1 in spSCC tumours (*Brabletz et al., 2017*) and precisely how the wounding event synergises with YAP to induce ZEB1 and EMT, as many other wound-induced signal transduction pathways (MRTF-SRF, Ras-AP1, TGFbeta-SMAD) have been implicated in EMT and have binding sites in the *ZEB1* regulatory region (*Bakiri et al., 2015*; *Brabletz et al., 2018*; *David et al., 2016*; *Davies et al., 2005*; *Gasparics and Sebe, 2018*; *Räsänen and Vaheri, 2010*; *Yang and Weinberg, 2008*). Overall, our findings begin to provide a molecular understanding of how genetic changes lead to SCC tumour growth and subsequent transformation to spSCC, and explain why exposure of skin to trauma, burns or ionizing radiation are so commonly associated with spSCC formation in patients (*McKee, 1996*).

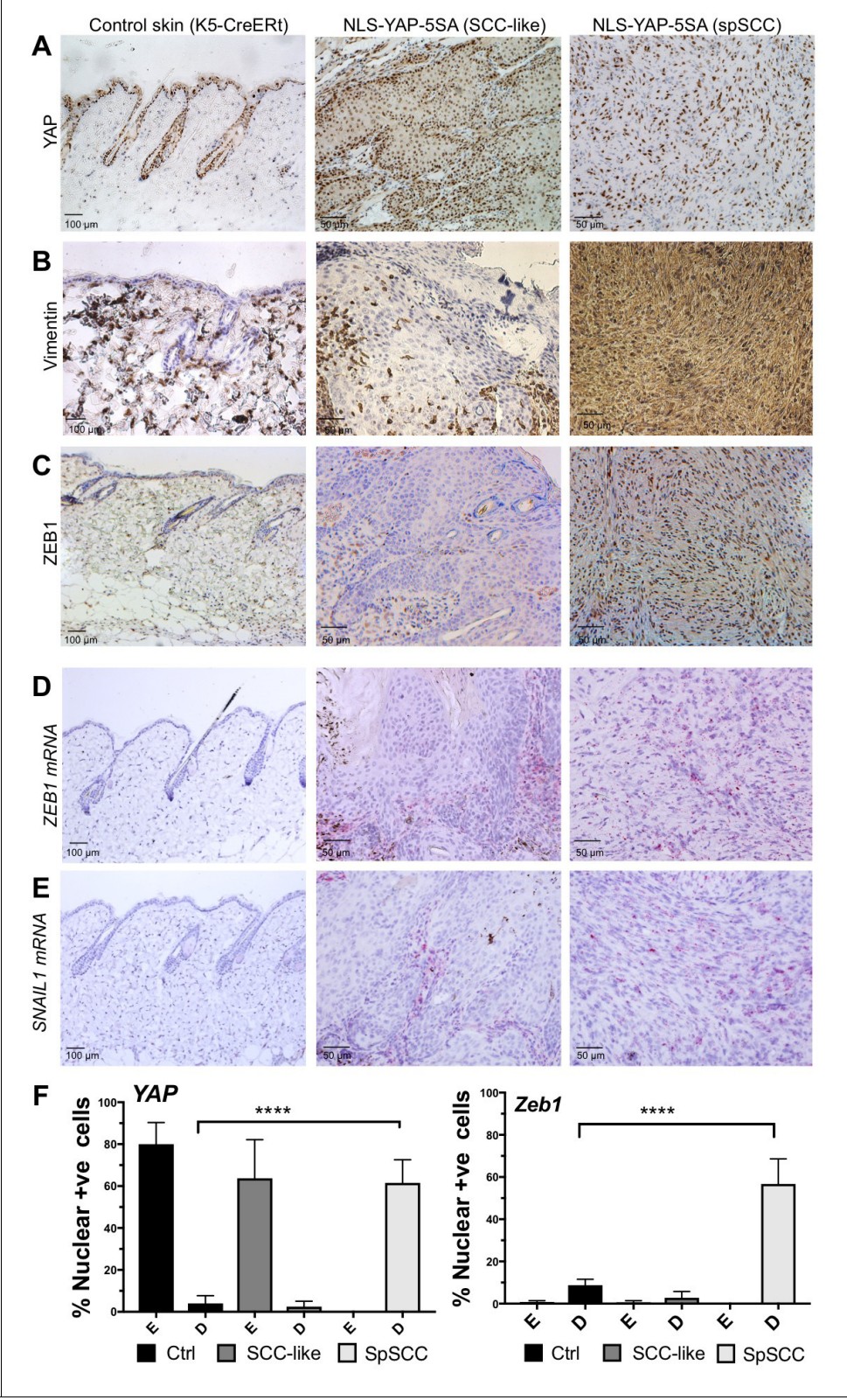

**Figure 3.** YAP-driven mouse spSCC formation involves transcriptional induction of ZEB1 expression and EMT. (**A**) YAP immunostaining of control skin as well as NLS-YAP-5SA driven SCC and spSCC tumours (*n* = 32). (**B**) Vimentin (mesenchymal marker) immunostaining of control skin as well as NLS-YAP-5SA driven SCC and spSCC tumours. Note strong induction in spSCC (*n* = 30). (**C**) ZEB1 (EMT transcription factor) immunostaining of control skin as well as NLS-YAP-5SA driven SCC and spSCC tumours. Note strong induction in spSCC (*n* = 33). (**D**) *ZEB1* mRNA in situ hybridisation of control skin
*Figure 3 continued on next page*

*Figure 3 continued*

as well as NLS-YAP-5SA driven SCC and spSCC tumours. Note strong induction in spSCC (*n* = 29). (**E**) *SNAIL1* mRNA in situ hybridisation of control skin as well as NLS-YAP-5SA driven SCC and spSCC tumours. Note strong induction in spSCC (*n* = 8). (**F**) Quantitation of YAP and ZEB1 marker expression in samples from wild-type skin, SCC-like, and spSCC-like mouse skin tumours (n > 30 samples for each case). Scale bars 50–100 μM.

DOI: https://doi.org/10.7554/eLife.33304.005

# Materials and methods

## Key resources table

| Reagent type (species) or resource | Designation | Source or reference | Identifiers | Additional information |
|---|---|---|---|---|
| Cell Line (Human) | HaCAT | Cell Services (Francis Crick Institute) Pubmed ID: 26989177 | (CLS Cat# 300493/p800_HaCaT, RRID:CVCL_0038) | |
| Antibody (Rabbit monoclonal) | anti-Vimentin | Abcam | (Abcam Cat# ab92547, RRID:AB_10562134) | 1/600 IHC |
| Antibody (Rabbit polyclonal) | anti-ZEB1 | Proteintech | (Proteintech Group Cat# 21544–1-AP, RRID:AB_10734325) | 1/500 IHC/1/100 IF |
| Antibody (Rabbit monoclonal) | anti-Keratin-5 | Abcam | (Abcam Cat# ab52635, RRID:AB_869890) | 1/500 IHC |
| Antibody (Rabbit polyclonal) | anti-beta-galactosidase | Acris | (Acris Antibodies GmbH Cat# R1064P, RRID:AB_973264) | 1/5000 IHC |
| Antibody (Rabbit polyclonal) | anti-E-Cadherin | Santa Cruz | (Santa Cruz Biotechnology Cat# sc-7870, RRID:AB_2076666) | 1/75 IHC |
| Antibody (Rabbit monoclonal) | anti-YAP | Cell Signalling Technology | (Cell Signaling Technology Cat# 14074, RRID:AB_2650491) | 1/400 O/N IHC |
| Antibody (Rabbit monoclonal) | anti-Ki67 | Abcam | (Abcam Cat# ab16667, RRID:AB_302459) | 1/350 IHC |
| Antibody (Mouse monoclonal) | anti-TEAD-1 | BD Biosciences | (BD Biosciences Cat# 610922, RRID:AB_398237) | 12.5 per 200 ug chromatin input CHIP |
| Antibody (Mouse monoclonal) | anti-YAP | Santa Cruz | (Santa Cruz Biotechnology Cat# sc-101199, RRID:AB_1131430) | 1/100 IF |
| Transfection reagent | Lipofectamine RNAiMAX | Thermo Fisher | Cat no: 13778075 | |
| siRNA | TEAD 1 | Dhamacon | Cat no: M-012603-01-0005 | 80 nM Final |
| siRNA | TEAD 2 | Dhamacon | M-012611-00-0005 | 80 nM Final |
| siRNA | TEAD 3 | Dhamacon | M-012604-01-0005 | 80 nM Final |
| siRNA | TEAD 4 | Dhamacon | M-019570-03-0005 | 80 nM Final |
| siRNA | YAP | Dhamacon | M-012200-00-0005 | 80 nM Final |
| Human Protein Atlas | Various | Pubmed ID: 16774037 | https://www.proteinatlas.org/ | |
| Human cancer samples | Vimentin/YAP | University of Southamption/Gareth Thomas | | |
| Chemical compound, drug | Dasatinib | Selleck Biochem | S1021 | 5 uM Final |
| Mouse strain | Rosa26-YAP5SA | Junhao Mao (University of Massachusetts Medical School) | | mixed background |
| Mouse strain | K5-CreERT2 | Ian Rosewell (Francis Crick Institute) | | mixed background |
| Mouse strain | Yapfl/fl Tazfl/fl | Axel Behrens (Francis Crick Institute) | | mixed background |

*Continued on next page*

*Continued*

| Reagent type (species) or resource | Designation | Source or reference | Identifiers | Additional information |
|---|---|---|---|---|
| Chemical compound, drug | 4-Hydroxytamoxifen | Sigma | H7904 | topical application of 200 ul oF 1.0 mg per 0.1 mL 4'OHT in DMSO on dorsal skin 5x consecutive days |
| Chemical compound, drug | Tamoxifen | Sigma | T5648 | IP 5 ul/g body weight of a 20 mg/ml solution in corn oil 5x consecutive days |
| RNA target probe | RNAscope Probe - Mm-Zeb1 | ACD | 451201 | |
| RNA target probe | RNAscope Probe - Mm-Snai1 | ACD | 451211 | |
| RNA target probe | RNAscope Probe - Mm-Snai2 | ACD | 451191 | |

## Cell culture

Human HaCAT cells (Francis Crick Institute cell services) were grown in DMEM (Gibco 41966) with 10% FCS and Penicillin/Streptomycin. All cells are subject to mycoplasma testing.

## Cell culture scratch assay

Cells are transfected with siRNA as previously described (Ref 13), before being replated at high density and left for 24 hr. Cells are scratched with a P200 pipette tip and left for 4 hr before being fixed. Cells treated with Dasatinib were treated and scratched simultaneously before being fixed. Cells were fixed with 4% PFA for 15 min before permeabilsing as previously described (Ref 13).

## Cell culture antibodies, image acquisition and quantification

Primary antibodies used were: Rabbit Zeb1 Proteintech 21544-AP) 1/100 and Mouse YAP (Santa cruz sc-101199) 1/100. Secondary antibodies were from Invitrogen, and used at 1:500 for 2 hr at room temperature along with DAPI. Cell culture samples were imaged with a Leica SP5 confocal microscope using a 63x oil immersion objective and processed using Adobe Photoshop. Cells were assessed over three independent experiments counting 200–300 cells per condition.

## siRNA transfection and inhibitor treatments

siRNA transfection experiments and inhibitor treatments were performed as previously described (Ref *Dong et al., 2007*).

Plasmids pCMV6 AC GFP ZEB-1 Transcript 1 (Origene) was transfected using Lipofectamine 3000 (Invitrogen). Quantification of extruded cells was performed by analysing n342 transfected cells over three experiments and recording number of extruded cells versus cells present within the monolayer.

## Chromatin immunoprecipitation

ChIP was performed as previously described (*Coda et al., 2017*), with the exception of the use of the ChIP Clean and Concentrate kit (Zymo Research, USA) for clean up of enriched chromatin according to the manufacturer's instructions. Quantitative PCR for enrichment at the *ZEB1* promoter was performed using the following primers: Fwd: 5'-GATGGGGAAGTGAGACAAGC-3'; Rev: 5'-CAGCTGGGATTGAAAGAGAGGC-3'.

## Animal procedures

### Ethics statement

All animal-regulated procedures were carried out according to Project License constraints (PPL 70/7926) and Home Office guidelines and regulations.

All experiments were carried out in accordance with the United Kingdom Animal Scientific Procedures Act (1986).

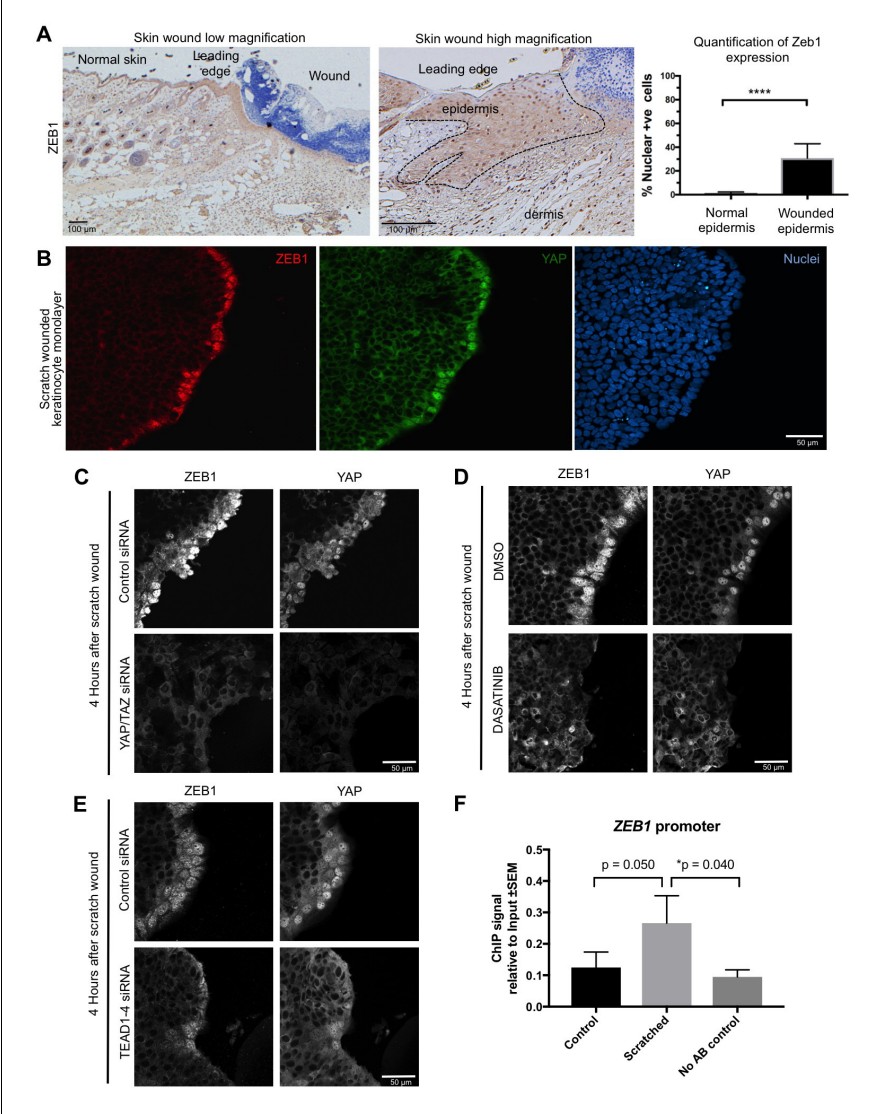

**Figure 4.** YAP promotes ZEB1 expression after epidermal wounding to drive EMT. (**A**) Punch wounding of mouse skin induces ZEB1 immunostaining in some leading edge cells. Scale bars 100 µM. (**B**) Scratch wounding of skin keratinocytes in culture induces ZEB1 and YAP immunostaining in leading edge cells. Scale bar 50 µM. (**C**) Induction of ZEB1 at the leading edge is prevented by transfection with siRNAs against YAP/TAZ. Scale bar 50 µM. (**D**) Induction of ZEB1 at the leading edge is prevented by treatment with Dasatinib, a Src-family kinase inhibitor that prevents YAP activation. Scale bar 50 µM. (**E**) Induction of ZEB1 at the leading edge is prevented by treatment with siRNAs against TEAD1-4. Scale bar 50 µM. (**F**) Chromatin Immunoprecipitation of TEAD1 at an upstream enhancer of the *ZEB1* gene in keratinocytes before or after scratch wounding. The weak enrichment may be caused by the small percentage of *ZEB1*-expressing cells in this experiment. Data were analysed by a Mann-Whitney Test n = 9 samples per experimental condition.

DOI: https://doi.org/10.7554/eLife.33304.006

## pROSA26-YAP5SA targeting construct

Diagram of the construct is shown in *Figure 2*. K5-CreERt mice were in mixed background. ROSA26-YAP5SA-NLS K5-CreERt mice were used with littermate controls.

## Mouse experiments

### R26-YAP5SA-NLS transgene expression in adult epidermis

Tamoxifen (Sigma, 20 mg/ml in con oil) was injected intraperitoneally (IP) (5 µl/g body weight) for five consecutive days into 8–16 week old controls or transgenic animals carrying K5-CreERt Yap R26-

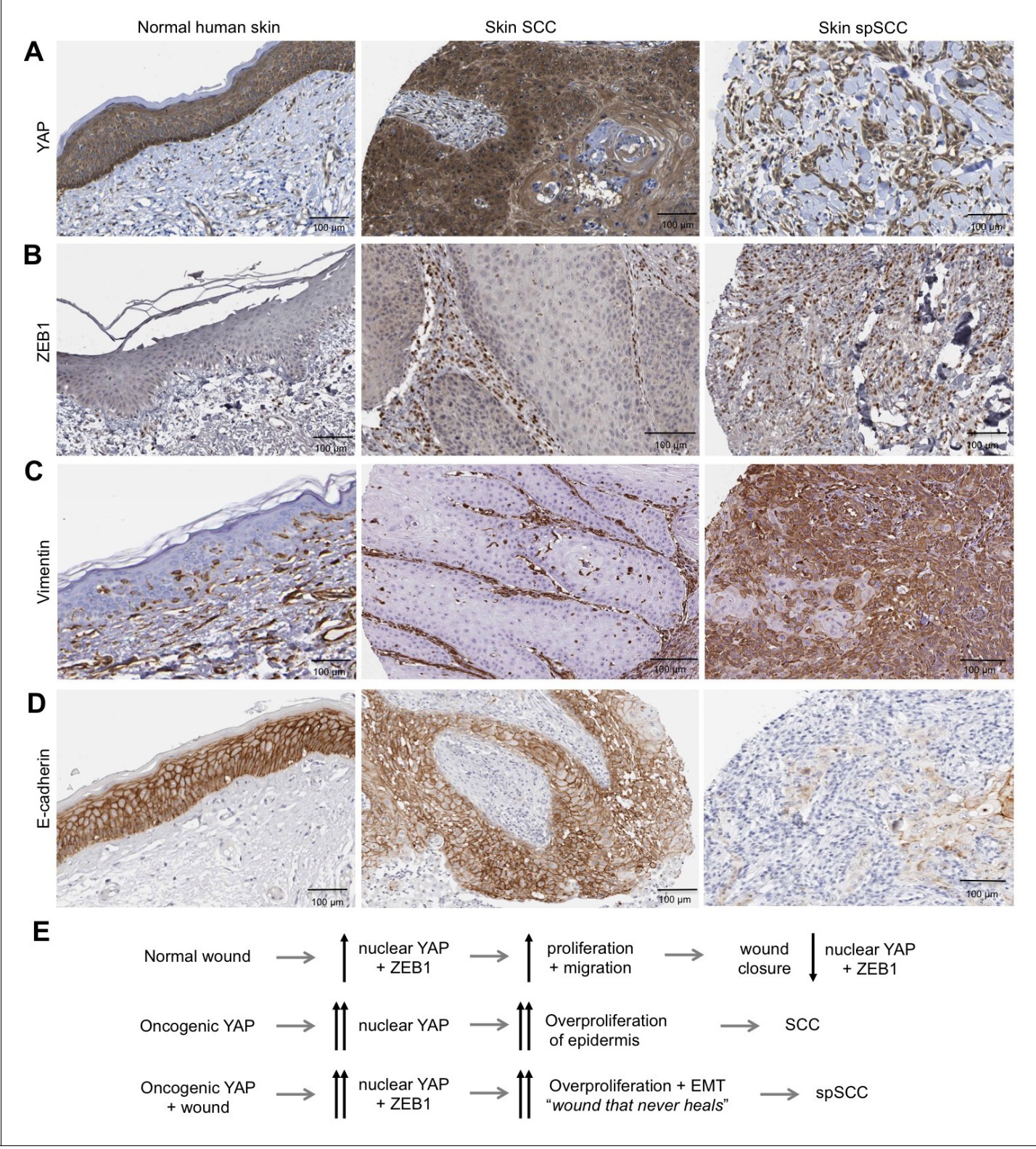

**Figure 5.** Human spSCC is characterised by co-expression of YAP and ZEB1. (**A**) YAP immunostaining of normal human skin, SCC and spSCC-like tumours. Note strong nuclear localisation in spindle-shaped spSCC tumour cells. (**B**) ZEB1 immunostaining of normal human skin, SCC and spSCC-like tumours. Note strong expression in spindle-shaped spSCC tumour cells. (**C**) Vimentin immunostaining of normal human skin, SCC and spSCC-like tumours. Note strong expression in spindle-shaped spSCC tumour cells. (**D**) E-cadherin immunostaining of normal human skin, SCC and spSCC-like

*Figure 5 continued on next page*

*Figure 5 continued*

tumours. Note absence of expression in spindle-shaped spSCC tumour cells. (E) Model comparing normal wound healing with SCC and spSCC formation. Scale bars 100 μM.

DOI: https://doi.org/10.7554/eLife.33304.007

The following figure supplement is available for figure 5:

**Figure supplement 1.** A panel of human spSCC tumours are characterised by widespread nuclear Zeb1 localisation.

DOI: https://doi.org/10.7554/eLife.33304.008

YAP5SA-NLS to induce YAP5SA-NLS expression and analyzed for LacZ (lineage tracer) expression by immunohistochemistry from 3 days thereafter. K5-CreERt R26-YAP5SA-NLS mice used for tumour formation analysis were analyzed by immunohistochemistry from 10 days after the initial tamoxifen treatment start.

## Wound healing

Following the 5 day tamoxifen treatment, four hydroxy-tamoxifen (4OHT, sigma) was topically applied to shaved backskin for five consecutive days at a dosage of 10 mg/ml in Ethanol and 100 μl was applied per mouse. Mice were anaesthetized with IsoFlo (Isoflurane, Abbott Animal Health) and treated with the analgesics Vetergesic (Alstoe Animal Health) and Rimadyl (Pfizer Animal Health) for 2 days after wounding. A 4 mm punch wound was made in the backskin using a biopsy punch (Miltex) and mice were culled 48 hr later, with the wound section harvested and fixed immediately for immunohistochemical analysis.

## Immunohistochemistry

Mouse backskin samples were harvested and fixed in neutral-buffered formaldehyde 10% vol/vol (sigma) and then embedded in paraffin in a head to tail orientation. The tissues were processed, embedded and sectioned at 4 μm and used for hematoxylin–eosin staining and immunohistochemistry. Sections were de-waxed in xylene, dehydrated by passage through graded alcohols to water. If required for antigen retrieval, sections were microwaved in citrate buffer pH6 for 15 min and then transferred to PBS. Endogenous peroxidase was blocked using 1.6% hydrogen peroxide in PBS for 10 min followed by washing in distilled water. Species specific blocking serum (Diluted to 10% in 1% BSA) was used to block non-specific staining in the tissue for 30 min. Slides were incubated with Primary antibody diluted to 1:100 in 1% BSA for 1 hr at room temperature. Sections were washed in PBS prior to applying the appropriate biotinylated secondary antibody for 45 min at room temperature. Sections were then washed in PBS and then incubated in ABC (Vector Laboratories PK-6100) for 30 min. Following washing in PBS, DAB solution was applied for 2–5 min with development of the colour reaction being monitored microscopically. Slides were washed in tap water, stained with a light haematoxylin, dehydrated, cleared and then mounted. Antibodies used for IHC were: Vimentin (Abcam ab92547) 1/600, Zeb1 (Proteintech 21544-AP) 1/500, Keratin-5 Abcam ab52635) 1/500, LacZ (Acris R1064P) 1/5000, E-Cadherin (Santa Cruz sc-7870) 1/75 O/N, YAP Cell signalling (14074) 1/400 O/N; (Santa cruz sc-101199) 1/200 O/N, Ki67 (Abcam ab16667) 1/350. Images were acquired with a Zeiss light microscope using 40x and 20x objectives. Additional images of human samples were obtained by data-mining the proteinatlas.org database.

## RNAscope (ACDbio)

Zeb1 probe was used according to manufacturer's instructions.

## Additional information

### Funding

| Funder | Grant reference number | Author |
| --- | --- | --- |
| Wellcome | FC001180 | Barry Thompson |
| Francis Crick Institute | FC001180 | Barry Thompson |

The funders had no role in study design, data collection and interpretation, or the decision to submit the work for publication.

## Author contributions
Zoé Vincent-Mistiaen, Investigation, Performed the mouse experiments, Designed research, Planned experiments, Analysed the data, Assisted with preparing the manuscript; Ahmed Elbediwy, Investigation, Performed the mammalian cell culture experiments, Performed the CHIP experiment, Designed research, Planned experiments, Analysed the data, Assisted with preparing the manuscript; Hannah Vanyai, Investigation, Methodology, Performed the CHIP experiment; Jennifer Cotton, Junhao Mao, Investigation, Developed and supplied the YAP 5SA NLS mice; Gordon Stamp, Investigation, Analysed the samples; Emma Nye, Bradley Spencer-Dene, Investigation, Stained and prepared the tissue sections; Gareth J Thomas, Data curation, Formal analysis, Investigation, Methodology, Provided the cancer tumour samples; Barry Thompson, Conceptualization, Supervision, Funding acquisition, Writing—original draft, Project administration, Writing—review and editing, Designed research, Planned experiments, Analysed the data, Wrote the manuscript

## Author ORCIDs
Ahmed Elbediwy (iD) http://orcid.org/0000-0002-2102-7339
Barry Thompson (iD) http://orcid.org/0000-0002-0103-040X

## Ethics
Animal experimentation: All animal-regulated procedures were carried out according to Project License constraints (PPL 70/7926) and Home Office guidelines and regulations. All experiments were carried out in accordance with the United Kingdom Animal Scientific Procedures Act (1986).

## Decision letter and Author response
Decision letter https://doi.org/10.7554/eLife.33304.012
Author response https://doi.org/10.7554/eLife.33304.013

## Additional files

### Supplementary files
• Transparent reporting form
DOI: https://doi.org/10.7554/eLife.33304.009

### Data availability
All data generated or analysed during this study are included in the manuscript and supporting files.

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
