## [Decision Letter]

Thank you for submitting your article "YAP drives cutaneous squamous cell carcinoma formation and progression" for consideration by *eLife*. Your article has been reviewed by three peer reviewers, one of whom is a member of our Board of Reviewing Editors and the evaluation has been overseen by Fiona Watt as the Senior Editor. The following individuals involved in review of your submission have agreed to reveal their identity: Eduardo Moreno (Reviewer #3).

The reviewers have discussed the reviews with one another and the Reviewing Editor has drafted this letter to draw your attention to significant concerns that you would need to address before this work could be considered for publication. Given the nature of the concerns, we request that you respond with proposed revisions and additional experimental work and an estimate of the time it would take to complete the essential changes. The editor and reviewers will then consider your proposal and issue a binding recommendation.

All reviewers thought this was a potentially interesting manuscript, but some experiments were a little preliminary at this stage. The conclusion that transgenic overexpression of constitutively nuclear and active YAP can drive spSCC formation was considered to be interesting. The authors also observed the formation of spindle cell carcinomas occurred at sites of wounding in their mice and this correlated with induction of Zeb1, although no direct mechanistic link was established. The authors attempted to correlate these observations in mice with studies of human cSCC and spindle cell carcinomas, although this requires further work (see below). Overall, the study, if improved, could provide evidence that nuclear YAP can drive spSCC formation on its own in K5-expressing cells at wound sites, and this may have potential relevance to human SCC.

Essential revisions:

1) The human data presented appear to be only from a small number of examples in Figure 1 and Figure 1—figure supplement 1. The authors should perform their analyses on larger sample sets and perform statistical and quantitative analysis of staining intensity and percentages of cells within skin and lesions of their markers of interest to determine the generality of their claims.

2) It is an important observation that spSCC can originate from K5-positive cells in the mouse. However, the LacZ lineage tracing is not clear enough in Figure 2C. The spindle SCC appears to be stained brown with little blue evident as presented. Clearer lineage tracing data (better LacZ staining) is required to substantiate this point.

3) The majority of the manuscript relies on IHC without providing evidence of the specificity of the antibodies used under FFPE conditions? e.g. the proteins of interest should be stained with the antibodies used to illustrate specificity in FFPE +/- protein knockdown.

4) The cooperation of YAP with Zeb1 for spSCC following wounding is interesting, but the data presented are correlative at this stage. The authors should extend these studies and provide evidence that Zeb1 is required for YAP effects in spSCC cells and/or provide more mechanistic links between YAP and ZEB1 (e.g. how does YAP influence ZEB1)?

5) Is it possible for the authors to provide more information from their animal experiments indicating how many tumours per mouse were formed and over what time scale following induction of Cre, using frequency and/or Kaplan-Meier analyses?

6) Can the authors provide evidence that the tumours specifically occurring at wounding sites are spSCCs (and not ulcerated tumours) – they should have this data.

7) Could the authors comment on why they think wounding contributes to spindle carcinomas. Is it just that the whole process of developing SCC tumours speeded up by a wound environment, e.g. via TGFbeta (which is known to cause spindle cell SCC tumours) or other inflammatory chemokines that cause a rapid EMT. Is YAP and Zeb1 needed for cSCC? This should be discussed.

---

## [Author Response]

Essential revisions:1) The human data presented appear to be only from a small number of examples in Figure 1 and Figure 1—figure supplement 1. The authors should perform their analyses on larger sample sets and perform statistical and quantitative analysis of staining intensity and percentages of cells within skin and lesions of their markers of interest to determine the generality of their claims.

As suggested, we have examined a panel of more than 20 human tumour samples and performed quantification of the YAP and ZEB1 expression to confirm the generality of our findings. The new data are in Figure 1—figure supplement 1 and Figure 5—figure supplement 1.

2) It is an important observation that spSCC can originate from K5-positive cells in the mouse. However, the LacZ lineage tracing is not clear enough in Figure 2C. The spindle SCC appears to be stained brown with little blue evident as presented. Clearer lineage tracing data (better LacZ staining) is required to substantiate this point.

We apologise for any confusion caused here. In this experiment, the LacZ lineage is stained brown with anti-betaGal immunostaining, rather than blue (which is an eosin counterstain). Thus, the entire spSCC tumour is strongly betaGal positive, indicating that all tumour cells arise from the K5-positive lineage. The manuscript text and figure legend have been updated to clarify this point.

3) The majority of the manuscript relies on IHC without providing evidence of the specificity of the antibodies used under FFPE conditions? e.g. the proteins of interest should be stained with the antibodies used to illustrate specificity in FFPE +/- protein knockdown.

The IHC antibodies used have all been previously validated, and we apologise for not explicitly stating this fact in the main text. For example, the YAP antibody used was validated by immunostaining YAP knockout mouse skin in our previous manuscript (Elbediwy et al., 2016). An example of confirmation by immunostaining a knockout mouse is shown in Author response image 1 (wild-type left, YAP/TAZ double knockout on the right).

**Author response image 1. respfig1:** Validation of Rabbit anti-YAP antibody in knockout skin.

4) The cooperation of YAP with Zeb1 for spSCC following wounding is interesting, but the data presented are correlative at this stage. The authors should extend these studies and provide evidence that Zeb1 is required for YAP effects in spSCC cells and/or provide more mechanistic links between YAP and ZEB1 (e.g. how does YAP influence ZEB1)?

To provide more mechanistic insight into the YAP-ZEB1 cooperation following wounding, we tested whether the induction of *Zeb1* mRNA by YAP in keratinocytes is mediated by TEAD-dependent transcription. This experiment entailed siRNA knockdown of TEAD1-4, which strongly reduced expression of ZEB1 after wounding. Furthermore, we have performed Chromatin Immunoprecipitation (ChIP) of TEAD1 binding to its cognate recognition site in the *Zeb1* promoter before and after scratch wounding of skin keratinocytes. This correlates with our finding that ZEB1 induction follows wounding and suggests that this is caused, at least in part, by the binding of YAP/TEAD1 to the *Zeb1* promoter. Notably, the *Zeb1* gene is well established as being crucial for EMT in cell culture models and in pancreatic cancer. To demonstrate that *Zeb1* is also required for YAP-driven EMT in mice is outside the scope of the manuscript, as this experiment would require us to obtain *Zeb1 floxed* mice, cross them to the K5-CreERT nlsYAP5SA mice and breed them to homozygosity for *Zeb1 floxed*, before obtaining sufficient numbers of offspring to perform quantitative analysis of the effects in histological samples of the resulting tumours (at least one year or longer).

5) Is it possible for the authors to provide more information from their animal experiments indicating how many tumours per mouse were formed and over what time scale following induction of Cre, using frequency and/or Kaplan-Meier analyses?

Done. The data are now shown in Figure 2.

6) Can the authors provide evidence that the tumours specifically occurring at wounding sites are spSCCs (and not ulcerated tumours) – they should have this data.

Yes, we can confirm that we have always used histological analysis to define spSCCs and never rely on ulceration of the tumours as a proxy, as ulceration is also common in conventional SCC.

7) Could the authors comment on why they think wounding contributes to spindle carcinomas. Is it just that the whole process of developing SCC tumours speeded up by a wound environment, e.g. via TGFbeta (which is known to cause spindle cell SCC tumours) or other inflammatory chemokines that cause a rapid EMT. Is YAP and Zeb1 needed for cSCC? This should be discussed.

Yes, we now include commentary on this point in the Results and Discussion section. It is indeed a fascinating future question how the wounding event synergises with YAP to induce ZEB1 expression and rapid EMT. We have proposed specific hypotheses, for future testing, regarding wound-induced signals that could cooperate with YAP to activate ZEB1 expression, based on our understanding of the ZEB1 promoter and additional information. As requested, we have also included discussion of the fact that ZEB1 is specifically involved in spSCC and cannot be involved in conventional cutaneous SCC, as it is not normally expressed in either normal skin or skin SCCs. YAP is, of course, necessary for cSCC as reported by the Piccolo lab and, very recently, the Blanpain lab (citation added).